# Differential Expression of Genes between a Tolerant and a Susceptible Maize Line in Response to a *Sugarcane* *Mosaic Virus* Infection

**DOI:** 10.3390/v14081803

**Published:** 2022-08-17

**Authors:** Gustavo Rodríguez-Gómez, Pablo Vargas-Mejía, Laura Silva-Rosales

**Affiliations:** Laboratorio de Interacciones Planta-Virus, Departamento de Ingeniería Genética, Centro de Investigación y de Estudios Avanzados del IPN (Cinvestav), Unidad Irapuato Libramiento Norte Carretera Irapuato-León, Irapuato 36824, Mexico

**Keywords:** SCMV, RNAseq, *Hsp90-2*, ABC transporter, *eEF1α*, *ZmPiezo*, *ZmPVIP1*

## Abstract

To uncover novel genes associated with the *Sugarcane mosaic virus* (SCMV) response, we used RNA-Seq data to analyze differentially expressed genes (DEGs) and transcript expression pattern clusters between a tolerant/resistant (CI-RL1) and a susceptible (B73) line, in addition to the F1 progeny (CI-RL1xB73). A Gene Ontology (GO) enrichment of DEGs led us to propose three genes possibly associated with the CI-RL1 response: a heat shock 90-2 protein and two ABC transporters. Through a clustering analysis of the transcript expression patterns (CTEPs), we identified two genes putatively involved in viral systemic spread: the maize homologs to the PIEZO channel (*ZmPiezo*) and to the Potyvirus VPg Interacting Protein 1 (*ZmPVIP1*). We also observed the complex behavior of the maize eukaryotic factors *ZmeIF4E* and *Zm-elfa* (involved in translation), homologs to *eIF4E* and *eEF1α* in *A. thaliana*. Together, the DEG and CTEPs results lead us to suggest that the tolerant/resistant CI-RL1 response to the SCMV encompasses the action of diverse genes and, for the first time, that maize translation factors are associated with viral interaction.

## 1. Introduction

Maize is one of the most important foods and staple crops around the world. However, it is affected by a variety of pathogens, such as the *Sugarcane mosaic virus* (SCMV), an ssRNA virus member of the *Potyviridae* family that causes leaf mosaics, chlorosis, and stunting [1]. Additionally, the SCMV has been shown to establish synergistic infections, together with viruses from the *Machlomovirus* genus (*Tombusviridae* family), causing maize lethal necrosis (MLN), with devastating effects, resulting in losses of up to 90% in China and Africa [2,3]. The best strategy to cope with such viral infections is the use of resistant maize lines such as FAP1360A [4,5], Siyi [6], and TR42 [7]. The resistance of FAP1360A has been attributed to the presence of the *Scmv1* and *Scmv2* loci on chromosomes 3 and 6, respectively. *Scmv1* is an atypical thioredoxin h (Trx h) [8], whereas *Scmv2* is an auxin-binding protein (ABP) [8,9,10]. Another maize line named CI-RL1 was identified in inbred evaluations for resistance to potyviruses at CIMMYT [11]. Negative viral RNA strands of the SCMV isolate Veracruz 1 (SCMV-Ver 1) virus were found in the inoculated leaves of the CI-RL1 line but were absent in distal leaves [11]. Those observations led us to consider CI-RL1 as a resistant line, but it remains unclear whether it uses a *ZmTrx h*-based defense mechanism.

Plants defend against pathogens by relying on the innate immunity of each plant cell and on the systemic signals produced at the infected sites [12,13,14,15]. Plant innate immunity is activated after the recognition of microbial- or pathogen-associated molecular patterns (PAMPs and MAMPs) by transmembrane recognition receptors (PRRs) [14,15]. PAMP-triggered immunity (PTI) is general and non-specific, characterized by the activation of early responses such as the production of reactive oxygen species (ROS), signaling cascade inducers (i.e., mitogen-activated protein kinases, MAPKs), and the accumulation of callose [14,16]. Pathogens can synthesize molecules that interfere with the activation of PTI called effectors [17]. Plants can specifically recognize effectors if they possess resistant (R) genes, proteins with nucleotide-binding sites (NBSs), and leucine-rich repeat (LRR) domains [18], resulting in effector-triggered immunity (ETI) [15] and leading to the activation of ROS, MAPKs, and phytohormone signaling routes [19,20,21,22]. In addition, ETI also activates the production of defense proteins associated with cell death in an antiviral process called the hypersensitive response (HR) [23,24]. ETI relies on intracellular R genes to recognize avirulence (AVR) proteins derived from the RNA virus and triggers an HR or programed cell death (PCD) in resistant hosts [25,26]. On the other hand, the PTI response is regarded as RNA silencing [27,28], a mechanism triggered by the detection of double-stranded RNA (dsRNA). Upon detection, the dsRNA is cleaved by Dicer-like proteins (DCLs) into small interfering 21–25 ribonucleotides (sRNAs). One of the dsRNA strands is loaded in the RNA-induced silencing complex (RISC) by the Argonaute (AGO) proteins; the RNA strand is then used as a guide to direct RNA degradation [27]. Recent evidence suggests that the triggering of a classical PTI response by virus-derived nucleic acids is a PRR-recognized PAMP [26,29].

RNA viruses rely on interactions with specific host proteins to complete their replication cycle [30]. Mutations in these proteins lead to a resistant phenotype in the host. As these mutations are inherited recessively and the presence of the wild-type alleles results in susceptibility, these genes are called susceptibility factors [31]. Some of these host proteins are diverted and used for the translation of the viral genome or are recruited to enhance replication or to participate in virus movement [32]. Most of the susceptibility factors that yield potyvirus resistance correspond to alleles of the eukaryotic initiation factor 4E (*eIF4E*) [33]. The importance of *eIF4E* relies on its interaction with the potyvirus VPg protein, which is a key factor in the translation of the potyviral genome [34]. Another susceptibility factor is heat shock protein 70 (Hsp70), a chaperone interacting with the CP of *Potato virus A* (PVA)*,* necessary for viral gene expression and replication [35]. Two heat shock proteins, Hsp40 and Hsp90, are involved in the innate immunity of the plant [36,37,38,39]. Another susceptibility factor, plasma membrane-associated cation-binding protein 1 (PCap1), has been shown to be related to *Potato virus Y* (PVY) accumulation and cell-to-cell movement [40]. Concerning virus movement, the Potyvirus VPg-interacting protein (PVIP1) from *Pysum sativum* is associated with the long-distance movement of the *Pysum sativum mosaic virus* (PsMV) [41]. A homologous PVIP mutant in *Arabidopsis thaliana* (L.) Heynh. does not allow VPg interaction and restricts the long-distance movement of the *Turnip mosaic virus* (TuMV), thus granting resistance. The role of these genes in maize resistance to the SCMV has not been explored.

Plants also cope with pathogen infection by using another mechanism called tolerance. While resistance limits pathogen multiplication [42,43,44], tolerance reduces the effect of the infection on the host’s fitness, regardless of the level of pathogen multiplication [44,45,46]. Both mechanisms—resistance and tolerance—coexist, and there is no evidence that tolerance necessarily implies an increase in pathogen multiplication [44].

To better understand the CI-RL1 response to the SCMV, we analyzed differentially expressed genes among a tolerant line (CI-RL1), a susceptible line (B73), and the F1 progeny derived from both lines. We used two main approaches with transcriptomic analyses. The first involved the identification of the differentially expressed genes (DEGs) in the functional enrichment network of biological processes (BPs) and cellular components (CCs), whereas the second involved an analysis of the clustering of transcript expression patterns (CTEPs) to allocate previously reported candidate genes. Through our analyses of DEGs and the Gene Ontology (GO) enrichment of BPs and CCs, three candidates stood out: heat shock protein 90-2 and two ABC transporters. CTEPs provided additional evidence for the diverse (and hence complex) behavior of the *eIF4E* homologs in maize—particularly the *eEF1α* factors. CTEP analysis also allowed the identification of two genes implied in long-distance movement: the maize homologs to the PIEZO channel (*ZmPiezo*) and to the Potyvirus VPg Interacting Protein 1 (*ZmPVIP1*).

## 2. Materials and Methods

### 2.1. Virus and Plant Materials

A previously characterized SCMV isolate, SCMV-Ver1, was used as the viral inoculum and was kept at −70 °C. The inoculum was reactivated by inoculating B73 susceptible plants with a mixture of approximately 500 mg of frozen ground infected tissue, 1 mL of PBS 1X buffer (Gibco, Life Technologies, Grand Island, NY, USA), and a small amount of carborundum. Inoculated leaves showing classic mosaic symptoms were then ground and used to further inoculate healthy maize plants at the third true leaf stage.

Three maize lines were used in this study. Two of them, CI-RL1 and B73, were obtained and donated by the Centro Internacional de Mejoramiento de Maíz y Trigo (CIMMYT). Additionally, an F1 line (CI-RL1 × B73) was generated in the state of Jalisco (Vallarta), Mexico using CI-RL1 as the female and B73 as the pollen donor. The potyvirus resistance donor for CI-RL1 was derived from a diverse background unrelated to line B73.

### 2.2. Experimental Design and Sequencing

We grew four plants of each line (B73, F1, and CI-RL1) in a greenhouse until they reached the third true leaf stage. All of the fully developed leaves were inoculated with the SCMV-Ver1 isolate. The plants were observed daily and at 7 dpi, the SCMV Coat Protein (SCMV-CP) cistron was detected around the ligule zone in the inoculated leaves but not in the asymptomatic portions or in the young systemic leaves of the susceptible plants. At 17 dpi, when SCMV-CP cistrons and symptoms appeared in the youngest leaves of the susceptible B73 line, the leaves from all lines were collected and stored in an ultra-freezer at −70 °C. The total RNA was extracted from 100 mg of frozen tissue using TRIzol reagent (Thermo Fisher, Waltham, MA, USA) according to the manufacturer’s instructions. The concentration of the resulting total RNA was determined with a Nanodrop 2000 spectrophotometer (Thermo Fisher Scientific, Wilmington, DE, USA). To confirm viral infection in the inoculated plants, we extracted RNA to amplify the SCMV-CP cistron through RT-PCRs (Figure 1C). We prepared a duplicate of each sample from the mix of RNA of four plants with the same RNA concentration. The mRNA was purified from the RNA pools using Dynabeads (Invitrogen, Waltham, MA, USA) according to the manufacturer’s instructions. A total of 24 paired-end (2 × 150) RNA-Seq libraries (six treatments from inoculated and non-inoculated B73, CI-RL1, and F1) were prepared and sequenced at the Beijing Genomics Institute facility, using DNBSEQTM technology (BGI, CHN). The raw data are publicly available from the National Center for Biotechnology Information (NCBI), under BioProject accession number PRJNA846583.

### 2.3. Mapping and Differential Expression Analysis

The trimming of the adaptors and the cleaning of low-quality reads were conducted as described previously [47]. Mapping and expression quantification were conducted with Kallisto v.0.46.1, as described by Vargas-Mejia [48]. Reads that failed to map to the reference B73 transcriptome retrieved from PLAZA 4.5 monocots (https://bioinformatics.psb.ugent.be/plaza/versions/plaza_v4_5_monocots (accessed on 4 June 2021)) were assembled de novo with Trinity v.2.11.0 under standard parameters [49]. Non-mapped reads were re-mapped to the de novo assembled transcripts and quantified as described earlier. The differential expression analysis was performed with Sleuth v.0.30.0, as reported before [50], comparing each infected maize line with its mock treatment. Genes with *p*-values of ≤0.05 were considered differentially expressed genes (DEGs).

### 2.4. Construction of CTEPs

The raw data of all the lines were mapped and quantified using Kallisto v.0.46.1 as described by Vargas-Mejia [48]. Then, the crude data were inserted in the Clust v.1.7.0 software where it was processed under standard parameters with TPM normalization [51]. Five CTEPs (C0 to C4) were obtained according to the basal expression level (BEL) of transcripts in the cluster: C0, low for B73 and F1 while high for CI-RL1; C1, intermediate for F1, low for B73, and high for CI-RL1; C2, low for B73 and high for F1 and CI-RL1; C3, low for RL and high for B73 and F1; C4, low for F1 and CI-RL1 and high for B73 (Figure 2A). Then, the clusters were grouped according only to their BELs for CI-RL1: high in Group 1 (C0, C1, and C2), and low in Group 2 (C3 and C4).

### 2.5. Functional Enrichment

Gene annotation data for reference B73 transcripts were downloaded from PLAZA 4.5 monocots (https://bioinformatics.psb.ugent.be/plaza/versions/plaza_v4_5_monocots (accessed on 4 June 2021)). For the de novo assembled transcripts, annotation was carried out using the Trinotate pipeline [52]. Then, a functional enrichment analysis was performed using PANTHER v17.0 with Fisher’s exact test, considering a false discovery rate of ≤0.05. Enrichment results were summarized with REVIGO (http://revigo.irb.hr/ (accessed 17 March 2022)), with a small similarity of 0.5 and SimRel as the semantic similarity score [53].

### 2.6. Gene Expression Validation

To validate the bioinformatic results of gene expression, we carried out an RT-qPCR analysis of 10 candidate genes (Appendix A), using the same source of RNA for sequencing. Additionally, 6 genes were evaluated at 2, 7, 12, and 17 dpi in three pools of leaves from two plants of each line. The cDNA synthesis was carried out with the RevertAid minus H enzyme (Thermo Scientific, Vilnius, Lithuania). RT-qPCR analyses were performed in a CFX96 Real-time system (BioRad, Hercules, CA, USA) with Maxima SYBR Green qPCR/ROX qPCR Master Mix (2X) (Thermo Scientific, Vilnius, Lithuania). The relative expression was calculated using the 2 ^(−^^ΔΔCt)^ method and was normalized with β-tubulin as the housekeeping gene.

## 3. Results

### 3.1. Symptom Development in B73, CI-RL 1, and F1 Progeny

We observed the plants daily for the presence of mosaic symptoms after inoculation with the SCMV-Ver 1 isolate. The susceptible B73 plants showed mosaic symptoms on the ligule zone as soon as 7 dpi, with a strong chlorotic mosaic in the first- and second-youngest leaves. Symptoms in the CI-RL1 and F1 progeny (F1) plants were not detected at this time. After 17 dpi, the CI-RL1 plants were asymptomatic and almost identical to the non-infected controls. Meanwhile, in the F1 leaves, a slight mosaic was observed alongside the central vein and in the basal region (Figure 1B). The F1 plants were taller, which could possibly be attributed to a heterosis effect. The presence of the SCMV-CP cistron was absent from the CI-RL1 samples but was found in all eight plants of the B73 and F1 samples (Figure 1C). Furthermore, the average expression levels of SCMV-CP measured in three pools of two plants for each line at 2, 7, 12, and 17 dpi showed increasingly high levels in the B73 line, increased and decreased levels in the F1 progeny, and decreasingly low levels in CI-RL1 (Table 1). At 30 dpi, we noticed continuously increasing levels in all the lines, but CI-RL1 showed viral levels that were 100 and 50 times lower than those of B73 and F1, respectively (Table 1).

### 3.2. Gene Expression Profiling Is Similar between Susceptible B73 and F1 Progeny

To analyze the global host response to the SCMV infection, we obtained the transcriptomes of the B73, F1, and CI-RL1 maize lines at 17 dpi and their respective sets of mock-inoculated plants. A total of 24 libraries were paired-end sequenced (PE 2 × 100), each one producing at least 10 million reads. After the differential expression analysis, genes with *p*-values of ≤0.05 were considered DEGs. We found a total of 5424 DEGs in all the maize lines; 2604 corresponded to B73, 1553 to F1, and 2406 to CI-RL1. For B73, we found 1793 to be upregulated and 811 to be downregulated. For F1, 959 and 594 were up- and downregulated, respectively, and 1048 were upregulated and 1358 were downregulated in the CI-RL1 line. Only 92 genes were differentially expressed in all three lines, corresponding to 1.69% of the total DEGs (Figure 3A). The susceptible B73 line and the F1 line were more similar, sharing 450 and 157 of the respective up- and downregulated genes (Figure 3B,C). The CI-RL1 line shared only 32 and 22 of the upregulated and 23 and 49 of the downregulated DEGs with F1 and B73, respectively. These results indicate a higher transcriptome similarity between the B73 and the F1 lines than with CI-RL.

### 3.3. Up- and Downregulation of Enriched Cellular Components after SCMV Infection

We performed a functional enrichment analysis to estimate the transcriptomic responses of our three different maize lines after the SCMV-Ver1 infection. We detected 34 upregulated and 11 downregulated enriched CCs in three main clusters (Figure 4). For the upregulated components, seven groups shared connections and three were individual CCs (GO terms: “intracellular protein-containing complex”, “chromosomal region”, and “protein containing-complex”). For B73 alone, six enriched CCs formed two groups: one contained three related to “catalytic and transferase complexes” and the other contained two, namely, “nucleoplasm” and “nuclear lumen” (Figure 3A, blue). For CI-RL1 alone, we found only one group containing nine CCs (GO terms: “chloroplast”, “chloroplast envelope”, “plastid envelope”, “chloroplast membrane”, “chloroplast outer membrane”, “chloroplast thylakoid membrane”, “thylakoid”, “plastid thylakoid”, and “plastid”) (Figure 4A, green). For the B73 + F1 combination (Figure 4A, orange), 14 GO terms were enriched and separated into three groups. The first contained nine CCs associated with the ribosome, the second included four related to the “nucleus and organelle membranes”, and the third was a mix of one for B73 + F1 (GO term “membrane”) and two “anchored components of membrane” for F1 (Figure 4A, red). The last group included only two CCs enriched in F1 alone, namely, the “external encapsulating structure” and “cell wall”.

Of the 11 downregulated transcripts in the CCs (Figure 4B), we observed one connected group and one isolated CC (GO term: “cytoplasm”). For the F1 line alone, the “ribonucleotide complex” (Figure 4B, red) interacted with the “protein-containing complex” CC (Figure 4B, purple), which was enriched in all lines. For CI-RL1 alone, only the “nuclear protein-containing complex” was present in the group (Figure 4B, green). The rest of the group members were enriched for CI-RL1 + F1 and these were related to “organelle” GO terms (Figure 4B, pink).

### 3.4. Enriched Biological Process Shows the Predominance of Downregulated CI-RL1 Transcripts

We found 98 upregulated and 67 downregulated BPs. Most of the upregulated BPs were in the B73 line, followed by F1 and RL. The BPs of B73 (Figure 5A, blue) showed enrichment in the processes related to DNA organization, DNA damage response, and DNA metabolism; the regulation of the metabolic process; cell wall biogenesis; and the production of secondary metabolites. The tallest bars were found in the F1 line and these were related to protein and amide biosynthesis (Figure 5A, red). For CI-RL1, the upregulated enriched BPs were related to photosynthesis and the responses to radiation, light stimuli, and abiotic stimuli (Figure 5A, green). In contrast, the BPs of the downregulated transcripts were mainly enriched in CI-RL1. For B73, the downregulated BPs were related to phytohormones and their signaling (Figure 5B, blue), whereas those enriched in CI-RL1 were related to DNA and RNA organization, metabolic processing, cellular response to stimuli, and the production of secondary metabolism. The F1 progeny shared most of its downregulated enriched processes with CI-RL1 (Figure 5B, green and red bars). The only two exclusively enriched BPs for F1 were related to the cellular macromolecule metabolic process and the amide biosynthetic process.

### 3.5. Candidate Genes Associated with the CI-RL 1 Tolerance/Resistance Response to SCMV Were Selected with Different Criteria

We used three criteria to select candidate genes associated with tolerance. First, we selected the most upregulated transcripts of the CI-RL1 line; second, we selected the CI-RL1 upregulated transcripts derived from the BP and CC enrichment study; and third, we made selections based on the expression levels of the transcripts between the three lines. For the first criterion, we chose the transcripts with log_2_ FC > 2.9. We then used the Monocots PLAZA 4.5 database to assign a V3 identifier that could be associated with a gene model, as well as a description indicating the possible function determined by gene homology. We chose 175 transcripts, of which 62 (35.4%) had no assigned V3 identifier, 17 (9.7%) had a V3 identifier without a description, and 31 (17.7%) did not have any identifier or description. The most highly upregulated transcript for the CI-RL1 line corresponded to the Zm00001d035392 transcript, with a log_2_ FC of 5.9, described as a “cycloartenol synthase” but without a V3 identifier. The lack of information for the transcripts led us to search according to the second criterion. In Table 2, we list thirty-two of the candidates with a putative relationship with tolerance. These candidates include eukaryotic translation initiation factor 6 (Zm00001d008223), heat shock protein 90-2 (Zm00001d031332), argonaute 1 (Zm00001d011096), and some ABC transporters. These candidates were up- or downregulated in only one or two of the lines. To search for genes with differential levels of expression between the three lines, the complete list of the DEGs for the B73 and F1 downregulated genes was compared with the full list of the upregulated genes for CI-RL1. We chose the genes present in the three lines, and the 19 with the highest expression for CI-RL1 are listed in Table 3. We also listed 19 genes with a higher expression for B73 after comparing the B73 and F1 upregulated genes with the downregulated genes of CI-RL1 (Table 3). In this set of data, we highlight the AP2-EREBP transcription factor (Zm00001d021205), CBF3-like protein (Zm00001d006169), and a catalase (Zm00001d027511). On the other hand, pathogenesis-related protein 5 (PR5; Zm00001d031158) and leucoanthocyanidin dioxygenase (Zm00001d030548) were strongly upregulated in B73 and downregulated in CI-RL1 (Table 3).

### 3.6. Thirteen Selected Candidate Genes Have High Basal Expression in CI-RL1

An analysis of the DEGs highlighted genes with possible roles in SCMV tolerance, for instance, argonaute and *Hsp90-2*. However, this approach may have excluded recessive resistance genes or susceptibility factors with unchanged expression levels. In order to detect these types of genes, we grouped five clusters of the transcripts’ expression patterns (CTEPs) according to the basal expression levels (BELs) for CI-RL1 (Figure 2A). Group 1 had a high BEL for CI-RL1 and included clusters C0, C1, and C2. Group 2 had a low BEL for CI-RL1 and contained clusters C3 and C4. After reviewing the current knowledge pertaining to the resistance genes of maize to the SCMV and other potyviruses, we generated a list of 57 candidate genes [54,55] and found 43 of them in our CTEPs. There were 13 in Group 1; 27 in both Groups 1 and 2, which were discarded (because they had high and low BELs for CI-RL1); and only 3 were found exclusively in Group 2 (Inositol-requiring protein-1 B *(IRE1B*), Plasmodesmata-located protein (*PDLP*) in C4, and Poly (A)-binding protein 8 (*PBAP8*) in C3); see Figure 2B. Of the 13 candidates in Group 1, 2 of them were associated with long-distance movement (Zm*PVIP1* and *ZmPiezo*), 3 with cell-to-cell movement (β-1,3-glucanase (*BG3)*, *PCap1*, and myosin), and 7 with the initiation or enhancement of viral replication (Essential for poteXvirus Accumulation 1 (*EXA1*), Sm motif protein (*LSM1*), *Prunus persica* DEAD-box RNA helicase-like (*PpDDXL)*, Inositol-requiring protein-1 A (*IRE1A*), Rice dwarf virus multiplication 1 (*RIM1*), Chloroplast phosphoglycerate kinase (*Chl-PGK*), and Re initiation supporting protein (*RISP*)). One of the listed genes (S-adenosyl-L-methionine synthetase (Os*SAMS1*)) does not participate in the virus infection cycle. In Group 2, two candidates were involved in virus replication (*IRE1B* and *PBAP8*) and one in cell-to-cell movement (*PDLP*).

### 3.7. Differences in Translation and Elongation Factor Coding Transcripts in the CI-RL1 Line

The interaction of the potyvirus VPg protein with the translation initiation factor eIF4E and/or its isoforms plays a central role in the virus infectious cycle [56]. A search in the maizeGDB database revealed that maize has six genes annotated as eukaryotic initiation factors (Appendix A). The maize *eIF4E* homolog was found in the C4 cluster annotated as *eif6* with low BELs in CI-RL1 and F1, although maintaining high BELs in B73. Furthermore, two more eIFs were identified in cluster C3: one corresponding to the gene model GRMZM2G022019 and annotated as *eif-7,* and other corresponding to the gene model GRMZM2G113096 without annotation and assigned as Zm*eIF4E*.

Additionally, the eukaryotic elongation factors (eEFs), which also participate in protein synthesis, were present in all clusters (C0-C4). Furthermore, as observed in the CTEPs analysis, different gene models and transcripts of the eEFs were found depending on their genetic backgrounds (Table 4). A search in the MaizeGDB database showed 11 genes annotated as eEFs (Appendix A). Thus, the results of the response to the SCMV could be an effect of the variety of the kinds of transcripts favored in each line.

As *eEF1α* was not detected in the list of DEGs, homologs were searched in the set of genes not surpassing the threshold established (>1 or −1 for up- and downregulated genes, respectively). Four homologs of *eEF1α* (Table 3, gray rows) with differential expression levels were found: the first was Zm00001d037873, corresponding to GRMZM2G154218, annotated as *elfa3*, and upregulated in B73 and F1; the second was Zm00001d036904, corresponding to GRMZM2G343543, annotated as *elfa10*, and upregulated in B73; the third was Zm00001d037877, corresponding to GRMZM2G001327, annotated as *elfa12*, and upregulated in B73 and F1 but downregulated in CI-RL1; and the fourth was Zm00001d025100, corresponding to GRMZM2G060842, without annotation but designated as *Zm-elfa*, and downregulated in CI-RL1.

## 4. Discussion

### 4.1. Resistance or Tolerance of the CI-RL1 Line

Using RNAseq analysis, we explored the possible role of a set of selected genes associated with the CI-RL1 response to the SCMV through analyses of DEGs and derived CTEPs. Maize resistance to the SCMV was reported more than 30 years ago [57], and only two genes proposed for resistance have been recently discovered: thioredoxin, corresponding to the locus *Scmv1*, and nABP corresponding to *Scmv2*. We have previously shown that the long-distance movement of SCMV-Ver1 is impaired in CI-RL1 [11]. The CI-RL1 line was originally deemed virus resistant due to the lack of SCMV symptoms and the absence of CP (cistron and protein) signals in distal leaves [11]. However, the finding of SCMV-Ver1 transcripts in systemic leaves via RNAseq and qRT-PCR analyses, in addition to genes in the CTEP analysis associated with the viral cycle, suggests that CI-RL is more of a virus-tolerant than a virus-resistant line.

### 4.2. DEG Analysis Pinpoints Multi-Genic Tolerance

After the analysis of the DEGs, seventy-four genes were selected: thirty-five derived from the enrichment of the BPs and CCs, one from the most highly upregulated genes for CI-RL 1, and thirty-eight for the DEGs between lines. The analysis of the enrichment of the CI-RL1 BPs and CCs highlighted genes related to photosynthesis, chloroplasts, and thylakoid membranes. During potyviral infection, the formation of vesicles, such as viral replication complexes (VRCs), is essential. Inside these vesicles, host cellular components are recruited and used for virus replication [58,59]. Some potyviruses use the chloroplast and ER membranes [60,61] to produce VRCs with concomitant chlorophyll breakdown and leaf yellowing during severe chlorosis [62,63,64]. As CI-RL1 tolerates the SCMV, the increase in photosynthesis and chloroplast-related transcripts could be a response aiming to alleviate the effects of the formation of VRCs. Such effects include the SCMV’s characteristic chlorosis (absent in CI-RL1) and an increase in ROS species. We also observed an increase in genes involved in the detoxification of ROS in B73. The liberation of fragments or the content of some organelles could act in signaling under the attack of a pathogen in the so-called damage-associated molecular pattern (DAMP), generating an ROS burst response [65,66,67,68]. The over-accumulation of ROS species (also called oxidative stress) can damage and disrupt the functioning of cellular components [69,70]. An effect on the enzymes involved in ROS detoxification has been reported in *Phaseolus vulgaris* plants treated with hormones and infected with a non-lesion-forming isolate of the *White clover mosaic potexvirus* (WClMV) [71]. The catalase, glutathione reductase, and superoxide dismutase activities of *P. vulgaris* were reduced, whereas peroxidase activity was increased [71]. In our case, eleven peroxidases were upregulated and only three were downregulated in the B73 line (Table 2). Additionally, a catalase corresponding to the gene model GRMZM2G090568 was downregulated in the B73 line, emulating the observations regarding *P. vulgaris*. Thus, the upregulation of peroxidases and the downregulation of catalases appear to be the responses of susceptible lines to infection. Here, CI-RL1 did not show the upregulation of peroxidase transcripts, but the catalase codified by the GRMZM2G090568 gene model was upregulated. Although the evidence is limited, it is tempting to speculate an effective and transient response to preventing cellular damage through ROS detoxification. This appears to be a strategy for CI-RL1 tolerance to the SCMV as a consequence of an early response to infection.

Among the thirty-five candidates from the BPs and CCs, the chaperone heat shock protein 90-2 (Hsp90-2) and ABC transporters were especially noteworthy. The chaperones enable mis-folded or aggregated proteins to correctly fold [72,73] and are also involved in targeting proteins for degradation [74]. They are also activated in response to biotic and abiotic stresses. Chaperone-mediated protein folding was up- and downregulated in the B73 line (Figure 5B). Potyviruses recruit the Hsp70 chaperone in the formation of replication complexes [35,75]. Another pair of Hsps—Hsp40 and Hsp90—appear to be involved in viral infections [36,37,38]. Hsp90 activates various cytosolic R proteins, such as the N tobacco protein, helping in the pathogen defense response [39]. The silencing of *Hsp90* by means of a virus-induced gene silencing (VIGS) strategy in *N. benthamiana* caused the loss of resistance to *P. syringae*, the *Tobacco mosaic virus* (TMV), and the *Potato X virus* (PVX)*,* confirming the key role of Hsp90 against pathogens [76]. The increase in the expression levels of *Hsp90-2*, corroborated by the qRT-PCR analysis (Appendix A), suggests a possible association with the observed tolerance response in the CI-RL1 line.

A diversity of ATP-binding cassette (ABC) transporters was detected in the BPs’ DEG data. The ABC transporters move diverse, structurally unrelated components across the membrane and intervene in plant–pathogen interactions [77,78,79,80]. These transporters have at least two transmembrane domains (TMD) and two ABCs or nucleotide binding domains (NBDs) [81]. Based on TMD and NBD sequence homology, the mammalian ABC transporters are grouped into seven subfamilies (ABC A to ABC G) [82,83,84,85]. In addition to transport, a defense-related function has been described in plants. A wheat ABC transporter (*lr34* gene) has been reported as a multi-disease resistance gene (MDR) against fungal pathogens [86,87]. Furthermore, maize plants expressing the *Lr34* wheat gene showed resistance against two fungal pathogens [88]. The protein codified by the maize gene model GRMZM2G014282 was identified as the best homolog to the Lr34 protein. The GCN-type ABC transporter from *Lilium regale* E.H. Wilson has been shown to be involved not only in defense against fungi but also against viruses [80]. This pathogen resistance is not due simply to the presence of the ABC transporter, as discussed by Sun et al. (2016) [80], but rather a more complicated network of genes in which the transporter plays a pivotal role appears to be involved. In our data, seven transcripts (Zm00001d050259, Zm00001d043762, Zm00001d004442, Zm00001d008178, Zm00001d048329, Zm00001d024497, and Zm00001d025043)—of which only six have a V3 identifier (GRMZM2G091478, GRMZM5G808836, GRMZM2G167658, GRMZM2G085111, GRMZM2G361256, and GRMZM2G165757)—were found (Table 2). Only two transcripts, corresponding to GRMZM2G167658 and GRMZM2G085111, were upregulated in the CI-RL1 line. The other transcripts were downregulated in CI-RL1 and upregulated in the B73 line or downregulated in B73 (Table 2). It is tempting to speculate that the high levels of expression of GRMZM2G085111 and GRMZM2G167658 imply their role in the SCMV tolerance response. However, our data are not sufficient to completely support such a hypothesis. Overall, GRMZM2G085111, GRMZM2G167658, and the rest of the ABC transporters deserve further research.

We found *CAS* (cycloartenol synthase; Zm00001d035392) to be the most highly expressed gene for the CI-RL1 line without a V3 identifier, and therefore a gene model could not be assigned. We found two more *CASs* to be upregulated in CI-RL1, corresponding to transcripts Zm00001d008671 and Zm00001d035389, with log_2_ FC values of 4.5 and 3.5, respectively. The latter had the V3 gene model identifier GRMZM2G390429. CAS is responsible for phytosterol biosynthesis, using oxidosqualene as a substrate [89]. Phytosterols are key compounds involved in maintaining membranous structures [89,90]. Although a role for CAS in virus resistance has not been reported, two *CAS1-like* genes were found in the region of the *Scmv2* locus [91]. Its closeness to the resistance gene for *Scmv2*, in addition to its expression level in CI-RL1, makes *CAS* a promising candidate that requires further experimental work. The lack of information available for the Zm00001d035392 and Zm00001d008671 transcripts makes it difficult to determine whether they are homologs or different transcripts of the same gene model.

### 4.3. PR5 and LDOX: Two Putative Susceptibility Factors in B73?

From the thirty-eight candidates obtained after the DEG comparison between the lines (Table 3), two were especially notable in B73 due to their contrasting expression levels: the Zm00001d031158 and Zm00001d030548 transcripts. The former has the V3 identifier GRMZM2G402631 and is described as a *PR5* gene, playing a key role in SAR as one of the PR proteins. PR5 is one of the thaumatin-like proteins (TLPs) [87]. They accumulate in response to biotic or abiotic stress, have antifungal properties in some plants, and function together with some resistance genes [92,93]. In tobacco, an increase in the expression of a thaumatin homologous to that in plants infected with the TMV was observed [94]. An extremely high level (log_2_ FC = 7) of expression was found for *PR5* in B73 and was negatively regulated for CI-RL1 (Table 3). This leads us to formulate two hypotheses about PR5: (i) it interacts with the SCMV in the viral cycle and (ii) it is part of the tolerance response and becomes downregulated. We discarded the second hypothesis based on our qRT-PCR data (Appendix A), which showed that *PR5* expression increased exclusively in B73. The second candidate gene, leucoanthocyanidin dioxygenase (*LDOX*), corresponds to GRMZM2G162158, an enzyme catalyzing the conversion of leucoanthocyanidins into anthocyanidins in the anthocyanin pathway [95]. The production of anthocyanins is induced under stress conditions or pathogen attack, presumably protecting the plant against oxidative stress [96]. The production of anthocyanins in B73 might be a response to virus infection, hence explaining the high expression level of the involved gene. As with *PR5*, the response of *LDOX* appears to be exclusive to B73 and negatively regulated in the CI-RL1 line. Both could be considered susceptibility factors in maize viral infections.

### 4.4. Genes Involved in Virus Replication Found in CTEP Analyses

We found two candidates associated with virus replication (in Group 2, Figure 2): *PABP8* and *IRE1B*. Three (*PABP2*, *PABP4*, and *PABP8*) out of the eight isoforms reported in *Arabidopsis* showed increased protein and mRNA levels during TuMV infection, in addition to the interaction of PABP2 with the VPg and RdRp of TuMV. Furthermore, in *pabp2pabp4* and *pabp2pabp8* double mutants, a reduction in TuMV mRNA levels was observed [97]. Although homologs for the three AtPABPs were searched for, only *PABP8* was detected in Group 2 of the C3 cluster (Table in Figure 2B). As PABP8 is possibly implicated in virus accumulation, its participation in tolerance was discarded.

*IRE1B* and *IRE1A* homologs in *Arabidopsis* function redundantly in the splicing of *bZIP60* during the response to biotic and abiotic stresses, resulting in the activation of stress-related proteins. The infection of *A. thaliana* with TuMV promotes the splicing of *bZIP60*. The *ire1aire1b* double mutant inoculated with TuMV showed delayed symptoms and low virus accumulation [98]. Thus, both IRE1 genes appear to be susceptibility factors in *Arabidopsis*. In maize, *IRE1A* was found in Group 1, whereas *IRE1B* was found in Group 2. In maize, IRE1A and IRE1B might positively affect viral infection and would not contribute to CI-RL1 tolerance.

The candidate genes *EXA1*, *LSM1*, *PpDDXL*, *RIM1*, *Chl-PGK*, and *RISP* (Appendix A), with high BELs in the CI-RL1 line (Group 1), play roles in viral replication [55] and were therefore not considered to be associated with the CI-RL1 response to the SCMV.

### 4.5. Cell-to-Cell Movement-Related Genes

Viruses move from cell to cell via plasmodesmata (PD), which regulate the size exclusion limit (SEL) with callose accumulation and the participation of viral movement proteins (MPs). Potyviruses, not having a specialized MP, rely on the CI, CP, HC-Pro, and VPg proteins for cell-to-cell movement. These proteins induce the formation and movement of VRCs in association with CI, towards and through the PD via the cytoskeleton [99]. Therefore, proteins that modify or interact with the PD can be considered important candidates in cell-to-cell movement. We found three candidates in Group 1 (*BG3*, *PCap1*, and myosin) and *PDLP* in Group 2 (Table in Figure 2B). *BG3* encodes an enzyme that degrades callose deposited in the plasmodesmata [100]. Callose deposition and its degradation are controlled by two enzymes: callose synthase (CalS) and BG3. It has been proposed that callose deposition restricts the spread of pathogens [101,102,103]. We speculate that the oxidative stress imposed upon viral infection could lead to callose accumulation in the PD [104]. This accumulation requires the activation of *BG3* for its degradation; experimental support should confirm such an association.

The expression level of the second candidate, *PCap1*, in *Capsicum annum* L. has been related to PVY accumulation and cell-to-cell movement [40]. Furthermore, protoplasts with a *pcap1* loss-of-function mutation in *A. thaliana* accumulated high levels of TuMV, discounting its participation in virus replication [105]. Additionally, the interaction between PCap1 and the P3N-PIPO from TuMV appears to be involved in the localization of the CI complex at the PD [99,105]. As PCap1 favors a possible increase in cell-to-cell transport, its involvement in SCMV tolerance appears unlikely.

The third candidate was myosin, a motor protein involved in a variety of mobility processes. As such, the silencing of myosin XI-2 from *Nicotiana benthamiana* Domin has been shown to inhibit the movement of the TMV, but not that of PVX or the *Tomato bushy stunt virus* (TBSV) [106]. Apparently, as plants possess different genes that code for myosin, viruses interact selectively with them. This myosin maize homolog was found to have a high expression level in the CI-RL1 line, which could favor the SCMV accumulation and movement, thus not being involved in host tolerance.

The only candidate in Group 2 associated with cell-to-cell movement was a PDLP belonging to a family of type-I membrane proteins. These proteins travel along the secretory pathway to reach the plasma membrane inside the PD [107,108]. PDLPs interact with viral movement proteins (MPs), which are capable of assembling as tubules within the PD [107]. As potyviruses do not belong to the tubule-forming group of viruses, the role of PDLPs in SCMV infection remains uncertain.

### 4.6. Long-Distance Movement-Related Genes

Only two candidates related to long-distance movement were found (Group 1, Figure 2): the Piezo and PVIP homologs. The *ZmPiezo* gene was chosen from the maize genome as an ortholog to the *ESC1* gene from *Arabidopsis*. *ESC1* codes for an ion channel, PIEZO, that responds to mechanical stimuli [109]. The *esc1* mutant showed alteration in the long-distance movement of TuMV in *A. thaliana* [110]. The *ESC1* gene, also called *AtPiezo*, was transcriptionally induced by viral infection [110]. The *ZmPiezo* gene was found to have a high BEL in the CI-RL1 line. Not much is known about *ZmPiezo* in maize, making it difficult to associate its high BEL with SCMV tolerance.

Concerning PVIP1, the *pvip1* loss-of-function mutant in *A. thaliana* inoculated with TuMV resulted in a lack of viral symptoms in planta [41]. This could be interpreted as indicating a need for PVIP1 to ensure viral infection. Here, not only did the CI-RL1 line have this gene, but it was found at high BELs (Appendix A), meaning that its role in SCMV tolerance is uncertain.

### 4.7. The Presence of Diverse Transcripts for eIFs and eEFs Implies the Complex Regulation of Translation in SCMV Interaction with Maize Transcripts in Resistance

Of paramount importance in this work is the role that translation seems to play in CI-RL1 tolerance. This was not surprising, as several of these translation initiation factors have been well documented during the past decades, although not in maize viral infection. However, little is known regarding the role of elongation factors in viral infection. It appears that the expression level of *eIF4E* is important for the establishment and maintenance of viral infection. Plants that transiently overexpress a modified *eIF4E* showed increase resistance to PVY and a decrease in the expression level of the host gene [111].

The ortholog of *eIF4E* from *Arabidopsis* (AT4G18040) corresponds to the maize *eif-6* transcript (Zm00001d041682), one of the six genes annotated as eukaryotic initiation factors in the maizeGDB database (Appendix A). Transcript *eif-6*, within cluster C4 (Table 3 and Figure 2A), had low BELs in the CI-RL1 line and the F1 progeny. On the other hand, *eif-7* and *ZmeIF4E* had low BELs only in the CI-RL1 line. The best match in the *Arabidopsis* genome for the *eif-7* genes corresponds to the isoform of the eIF4E factor. The use of eIF4E or its isoform depends completely on the interaction of the host and the infecting virus. Both *eif-7* and *ZmeIF4E* are newly described alleles in maize associated with tolerance, for instance, restricting viral movement in the CI-RL1 line, as has been described for the *Tobacco etch virus* (TEV) in *A. thaliana* [112]. Both B73 and CI-RL1 exhibited different BELs of *ZmeIF4E* (Table 4), suggesting either differences in their coding or promoter regions.

The eukaryotic elongation factors (eEFs) are encoded by genes playing a central role in the elongation step of translation [113]. One of them, namely, eukaryotic elongation factor 1α (eEF1α), is a cytoplasmatic protein that delivers aminoacylated tRNAs to ribosomes during polypeptide elongation when bound to GTP [114,115] and plays a role in nuclear export, proteolysis, and apoptosis [116]. In human viral infections, eEF1α interacts with the Gag1 polyprotein of human immunodeficiency virus type 1 (HIV-1) [117], with the 3′ stem-loop region of the West Nile virus (WNV) [118], and with the nucleocapsid protein of the paramyxovirus respiratory syncytial virus [119]. In plants, eEF1α binds tRNA-like structures (TLSs) in the 3′ region of the *Turnip yellow mosaic virus* (TYMV), the TMV, and the *Brome mosaic virus* (BMV) [120,121]. The aminoacylation of the TLSs enhances virus protein translation and facilitates virus RNA encapsidation [122,123,124]. The interaction between the ER-localized P3 from the *Soybean mosaic virus* (SMV) and eEF1α is essential for virulence in the susceptible host [115]. Furthermore, *eEF1A* from *N. benthamiana* is a pro-viral factor required for the *Tomato spotted wilt virus disease* (TSWV) [125]. Maize possesses at least eleven eEFs, and nine are annotated in the MaizeGDB database as “*elongation factor α* (*elfα)”* (Appendix A). A variety of functional transcripts are associated with each of them; for example, *elfa1* has 15 transcripts associated with this gene model. Although the expression levels of the *elfa* homologs were below the selection criteria, they could help us to understand the CI-RL1 response to the SCMV. We speculate that the high expression levels of *elfa10*, *eIfa12*, and *elfa3* in the B73 and F1 progeny plants (Table 2, gray cells) suggest that these factors interact with the SCMV in a yet-unforeseen manner. In contrast, CI-RL1 showed downregulation in *elfa12* and *Zm-elfa* (Table 2, gray cells). It is possible that *eIfa3*, *elfa10,* and *eIfa12* interact more efficiently with the SCMV. Maintaining the low expression of *elfa12* and *Zm-elfa* in CI-RL1 aids in the tolerance response. Additionally, according to the data obtained in the CTEP analysis, different transcripts of the same gene model with high or low BELs could be found in the clusters (Table 4), suggesting the intricate regulation of these factors in maize during SCMV infection. It will be necessary to dissect the interaction of elfa10, elfa3, elfa12, and Zm-elfa with the SCMV to disentangle the myriad of translation factors involved in this host–virus interaction.

In addition to the large number of possibly important genes for resistance, SNPs also need to be considered for some of the candidate genes discussed here, adding even more complexity than anticipated in maize–SCMV interactions. Finding key SNPs will contribute to the development of breeding programs against devastating viral diseases in monocots. Furthermore, increasing maize genome annotations will enrich our understanding of tolerance/resistance in maize.

## 5. Conclusions

We performed an RNAseq analysis of a CI-RL1 line initially classified as resistant, a susceptible line B73, and the F1 progeny of both lines. The CI-RL1 line was originally designated as resistant due to the absence of SCMV symptoms and the lack of detection of the CP cistron and protein in distal leaves. However, the RNAseq analyses presented in this work led us to consider CI-RL1 as a tolerant line. A GO enrichment analysis of the CCs and BPs led to the identification of two ABC transporters and *Hsp90-2* as possible candidates in virus tolerance. We also found two genes implicated in virus long-distance movement—*ZmPiezo* and *ZmPVIP*—in maize. Additionally, *CAS* was identified using a different selection criterion (mostly expressed in CI-RL1 and DEGs between lines). This candidate appears to have a role in tolerance to the SCMV. *PR5* and *LDOX* seem to be susceptibility factors for the SCMV.

Using CTEP analysis, we observed complex behavior by the *eIF4E* homologs in maize, particularly the *eEF1α* factors. Furthermore, two new maize gene models—*ZmeIF4E* and *Zm-elfa*—were identified as possibly being involved in the SCMV interaction. The validity of CTEPs as a tool for uncovering susceptibility factors needs to be proved by sequencing and detecting SNPs for selected candidate genes.

More work is needed to understand how the genes identified in this work—*ZmeIF4E, Zm-elfa, ZmPiezo*, *PR5*, *CAS*, *LDOX*, *ZmPVIP*, *Hsp90-2*, and ABC transporters—are involved in maize potyvirus tolerance/resistance.

## Figures and Tables

**Figure 1 viruses-14-01803-f001:**
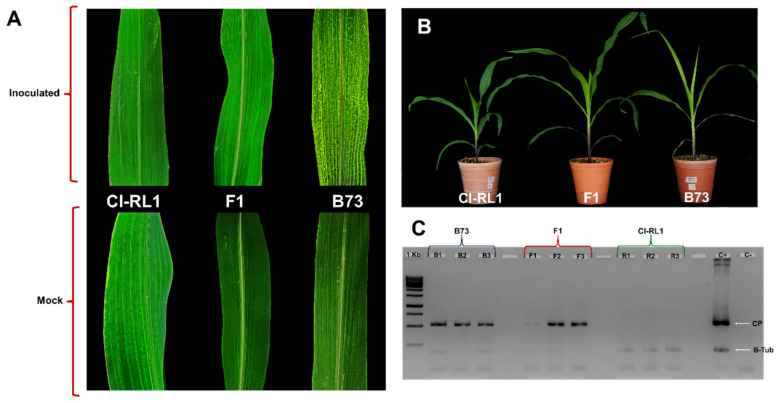
Symptom development and *Sugarcane mosaic virus* (SCMV) detection in three maize lines inoculated with SCMV isolated Veracruz 1 (SCMV-Ver1) in leaves at the same phenological stage: (**A**) B73 susceptible-symptomatic line, with chlorotic mosaic symptoms, and CI-RL1 and F1, without evident symptoms; (**B**) SCMV-Ver1-infected plants (CI-RL1, F1, and B73 from left to right). (**C**) Presence of the SCMV Coat Protein (CP)-cistron in three plants randomly selected from each line.

**Figure 2 viruses-14-01803-f002:**
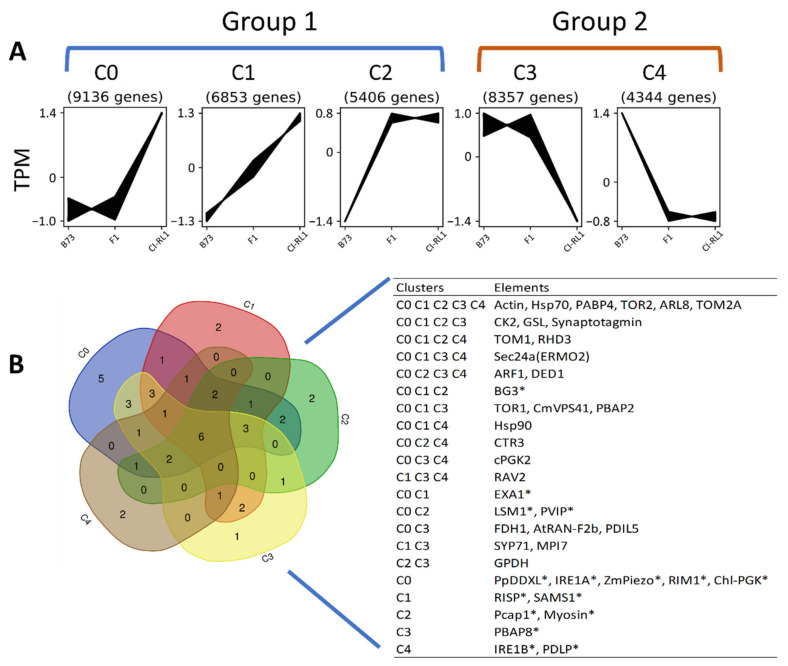
Cluster groups of transcript expression patterns (non-DEGs) in three maize lines: B73, F1 (B73 × CI-RL), and CI-RL1. (**A**) Group 1, with C0 to C2 clusters having high expression levels for CI-RL1; Group 2, including C3 and C4 clusters with low expression levels for CI-RL1. (**B**) Venn diagram of 43 selected candidate genes present in one or more expression pattern clusters. * Set of sixteen candidate genes found in only one group (Group 1 or 2).

**Figure 3 viruses-14-01803-f003:**
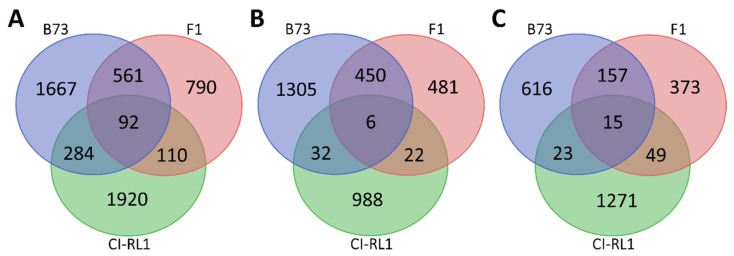
Venn diagrams of differentially expressed genes (DEGs), showing B73 susceptible (blue), F1 progeny (red), and CI-RL1 tolerant lines (green). (**A**) All DEGs (*p*-values ≤ 0.01); (**B**) upregulated and (**C**) downregulated DEGs.

**Figure 4 viruses-14-01803-f004:**
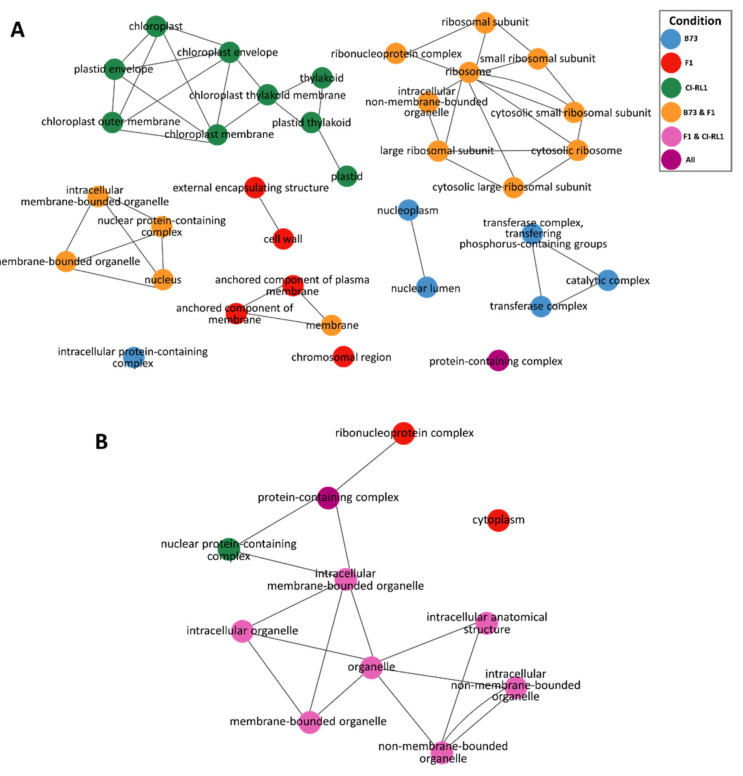
Functional enrichment network of cellular components (CCs) of differentially expressed genes (DEGs) in B73, CI-RL1, and F1 progeny. (**A**) Enrichment network of upregulated (**B**) and downregulated genes. Nodes represent an enriched CC ontology term, and edges represent the relationships between these ontologies. CC shared terms correspond to the following: B73 alone (blue); shared between B73 and F1 (orange); shared between all lines (purple); shared between CI-RL1 and F1 (pink); and those corresponding to CI-RL1 alone (green).

**Figure 5 viruses-14-01803-f005:**
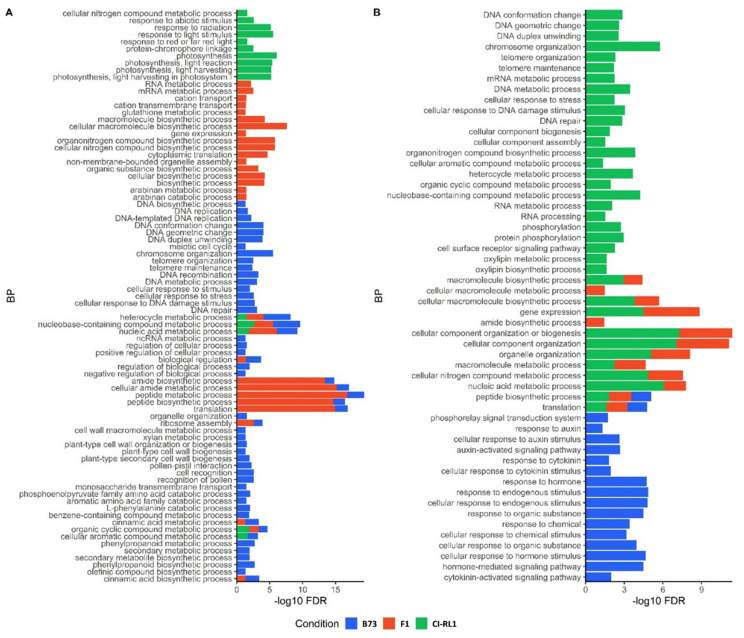
Bar plot of functionally up- (**A**) and downregulated (**B**) genes of the enriched biological processes (BPs) for B73 susceptible (blue), F1 progeny, (red), and CI-RL1 (in green) lines. (**A**) The *x*-axis represents the percentage of background annotated genes for each BP term.

**Table 1 viruses-14-01803-t001:** Relative expression levels of SCMV Coat Protein (SCMV-CP) in the three different lines at 2, 7, 12, 17, and 30 dpi measured via qRT-PCR.

Line	2 dpi	7 dpi	12 dpi	17 dpi	30 dpi
B73	2.2	34.3	7090.5	43,588.7	93,975.7
F1	5.0	0.9	57.5	0.2	32,616.9
CI-RL1	1.8	1.5	0.8	0.8	72.8

**Table 2 viruses-14-01803-t002:** Thirty-two candidate genes selected using different criteria. Down- (in blue) and upregulated (in red) genes, maize transcript IDs, V3 identifiers, fold-change expression levels (log_2_), and descriptions as reported in the Monocots PLAZA 4.5 database are presented. The candidates with a gray background were below the selection criteria threshold for DEGs.

Go Term	ID	V3 Identifier	log_2_ FC			Description
			B73	F1	CI-RL1	
Nucleic acid metabolic process	Zm00001d008223	GRMZM2G304121		−1.2	3.4	Eukaryotic translation initiation factor 6
	Zm00001d035823	GRMZM2G036525			2.9	Tetratricopeptide repeat (TPR)-like superfamily protein
	Zm00001d004415	GRMZM2G327247			1.6	RNA-dependent RNA polymerase
	Zm00001d026398	GRMZM2G445575	−1.3		1.4	Transcription factor TGA4
Response to abiotic stimulus	Zm00001d031332	GRMZM2G024668			2.0	Heat shock protein 90-2
	Zm00001d027511	GRMZM2G090568	−1.3	−1.2	2.0	Catalase/Catalase isozyme 2
	Zm00001d031325	GRMZM2G080724			1.9	25.3 kDa heat shock protein chloroplastic
Translation	Zm00001d034776	GRMZM2G040369	2.5	1.4		Elongation factor 2
Regulation of biological process	Zm00001d044242	GRMZM2G081816	2.1	1.6		Transcription factor bHLH87
Negative regulation of biological process	Zm00001d011096	GRMZM2G162525	1.7	1.4		Argonaute1
	Zm00001d050259	GRMZM2G091478	3.7		−1.5	ABC transporter A family member 7
	Zm00001d043762	GRMZM5G808836	1.9	1.3	−2.8	ABC transporter B family member 9
	Zm00001d004442	GRMZM2G167658	−2.1		1.7	ABC transporter B family member 19
	Zm00001d008178	GRMZM2G085111			1.0	ABC transporter B family member 21
	Zm00001d048329	GRMZM2G361256	1.8		−1.7	ABC transporter C family member 3
	Zm00001d025043		1.5		−1.2	ABC transporter C family member 4
	Zm00001d024497	GRMZM2G165757	−2.5			ABC transporter G family member 14
Cellular response to stress	Zm00001d053554	GRMZM2G117706	4.3			Peroxidase/Peroxidase 52
	Zm00001d052335		2.8			Peroxidase/Peroxidase 67
	Zm00001d040705	GRMZM2G107228	2.6		−2.2	Peroxidase/Peroxidase 64
	Zm00001d008173	GRMZM2G122853	1.8			Peroxidase/Peroxidase 2
	Zm00001d042022	GRMZM2G103342	1.6		−1.8	Peroxidase/Peroxidase 12
	Zm00001d014341	GRMZM2G150893	1.5	1.1		Peroxidase/Peroxidase 53
	Zm00001d040364	GRMZM2G089982	1.3			Peroxidase/Peroxidase 72
	Zm00001d007161	GRMZM2G450717	1.3			Peroxidase/Peroxidase 52
	Zm00001d038599	GRMZM2G089895	1.2			Peroxidase/Peroxidase 1
	Zm00001d029604	GRMZM2G020523	1.2			Peroxidase/Peroxidase 2
Response to light stimulus	Zm00001d027511	GRMZM2G090568	−1.3	−1.2	2.0	Catalase/Catalase isozyme 2
Response to chemical	Zm00001d037894	GRMZM2G079440	−4.8	−4.3		RAB17 protein/Dehydrin DHN1
Hormone-mediated signaling pathway	Zm00001d023659	GRMZM2G006042	−1.1			Auxin response factor 2
	Zm00001d018178	GRMZM2G479760	−1.0			ABSCISIC ACID-INSENSITIVE 5-like protein 5
Cellular nitrogen compound metabolic process	Zm00001d031953	GRMZM5G882446			−1.2	Glutathione S-transferase
Response to hormone	Zm00001d014606	GRMZM2G133434	−1.2			Peroxidase/Peroxidase 45
	Zm00001d020808	GRMZM5G843748	−1.1	−1.6		Peroxidase/Peroxidase 17
	Zm00001d009373	GRMZM2G023840	−1.5			Peroxidase/Peroxidase 72
	Zm00001d037873	GRMZM2G154218	0.8	0.9		Elongation factor 1-alpha (*elfa3*)
	Zm00001d036904	GRMZM2G343543		0.6		Elongation factor 1-alpha (*elfa10*)
	Zm00001d037877	GRMZM2G001327	0.7	0.7	−0.5	Elongation factor 1-alpha (*elfa12*)
	Zm00001d025100	GRMZM5G801457			−0.7	Translation elongation factor EF1A (*Zm-elfa*)

**Table 3 viruses-14-01803-t003:** Candidate genes selected after differentially expressed genes (DEGs) analysis. Down- (in blue) and upregulated (in red) genes, maize transcript IDs, fold-change expression levels (log_2_), and the information for V3 identifiers and descriptions are shown.

ID	V3 Identifier	log_2_ FC			Description
		B73	F1	CI-RL1	
Zm00001d021205	GRMZM2G069082	−2.0	−0.9	2.7	AP2-EREBP transcription factor/Dehydration-responsive element-binding protein 1B
Zm00001d006169	GRMZM2G124037	−3.0	−1.2	2.2	CBF3-like protein; CRT/DRE binding factor; DREB-like protein
Zm00001d021208	GRMZM2G069146	−2.0	−0.8	2.0	AP2-EREBP transcription factor/Dehydration-responsive element-binding protein 1B
Zm00001d027511	GRMZM2G090568	−1.3	−1.2	2.0	Catalase/Catalase isozyme 2
Zm00001d031745	GRMZM2G142962	−1.5	−0.6	1.5	Homeobox-leucine zipper protein HAT3
Zm00001d016361	GRMZM2G031983	−1.3	0.9	1.5	C2C2-GATA transcription factor/Putative GATA transcription factor 22
Zm00001d038850	GRMZM2G111017	−1.1	−0.7	1.3	Probable ADPATP carrier protein
Zm00001d053911	GRMZM2G042664	−1.2	−0.4	1.3	expressed protein
Zm00001d043060	GRMZM2G059562	−1.7	−1.1	1.3	Probable WRKY transcription factor 30
Zm00001d044117	GRMZM2G150260	−1.1	1.9	1.2	DNA binding protein
Zm00001d043505	GRMZM2G039246	−1.5	−1.4	1.2	Histidine-containing phosphotransfer protein 4
Zm00001d033510	GRMZM2G146616	−1.5	−0.5	1.2	expressed protein
Zm00001d018107	AC220927.3_FG007	−1.3	−0.9	1.2	Protein EXORDIUM
Zm00001d033786	GRMZM2G155767	−1.7	−0.8	1.2	histidine kinase4
Zm00001d016982	GRMZM2G176253	−1.4	−1.2	1.2	Protein NRT1/PTR FAMILY 6.4
Zm00001d025470	GRMZM2G158097	−1.2	−0.5	1.2	expressed protein
Zm00001d010490	GRMZM2G084489	−1.3	−0.6	1.1	CW-type Zinc Finger
Zm00001d002256	GRMZM5G897776	−1.5	−0.6	1.0	Starch synthase 3 chloroplastic/amyloplastic
Zm00001d019217	GRMZM2G031780	−1.0	−0.7	1.0	ABC2 homolog 13
Zm00001d031158	GRMZM2G402631	7.0	4.0	−2.7	Pathogenesis-related protein5
Zm00001d030548	GRMZM2G162158	6.0	2.8	−2.0	Leucoanthocyanidin dioxygenase
Zm00001d002707	GRMZM2G120016	4.8	5.2	−3.6	Glycosyltransferase/Cis-zeatin O-glucosyltransferase 1
Zm00001d052690	AC233883.1_FG006	3.8	2.2	−3.4	UPF0481 protein
Zm00001d049632	GRMZM2G007587	3.7	1.6	−1.4	Kelch motif family protein isoform 1\F-box/kelch-repeat protein
Zm00001d044904	GRMZM2G017244	3.6	2.5	−1.5	Transcription factor EMB1444
Zm00001d002436	GRMZM2G125669	3.2	2.8	−3.6	Ubiquinol oxidase/alternative oxidase2
Zm00001d048683	GRMZM2G106377	3.2	1.6	−2.1	Putative RING zinc finger domain superfamily protein
Zm00001d052989	GRMZM2G130389	3.1	1.4	−1.1	Cysteine-rich receptor-like protein kinase 6
Zm00001d024963	GRMZM2G330635	2.9	2.1	−1.7	Glutathione S-transferase GSTU6
Zm00001d032869	GRMZM5G896540	2.8	1.1	−2.0	L-type lectin-domain containing receptor kinase V.9
Zm00001d039077	GRMZM5G846057	2.8	0.9	−2.5	AP2-EREBP transcription factor/Ethylene-responsive transcription factor ERF061
Zm00001d005028	GRMZM2G018436	2.7	1.1	−2.0	NAC domain-containing protein 77
Zm00001d040743	GRMZM2G091088	2.7	1.1	−1.0	Os01g23380-like protein
Zm00001d044874	GRMZM2G135960	2.6	1.4	−4.0	23.6 kDa heat shock protein mitochondrial
Zm00001d048949	GRMZM2G117971	2.6	−2.6	−1.9	Hevein-like preproprotein
Zm00001d015623	GRMZM2G417945	2.5	1.1	−1.8	Glycosyltransferase/Limonoid UDP-glucosyltransferase
Zm00001d034501	GRMZM2G137321	2.5	1.1	−2.7	AAA-ATPase ASD mitochondrial
Zm00001d038451	GRMZM2G169966	2.4	1.0	−2.7	Putative WRKY DNA-binding domain superfamily protein

**Table 4 viruses-14-01803-t004:** Transcripts coding for translation/elongation factors found in clustering of transcript expression patterns (CTEPs) C0 to C4 (see Figure 2A).

C0		C1		C2		C3		C4	
Gene	Transcript ID	Gene	Transcript ID	Gene	Transcript ID	Gene	Transcript ID	Gene	Transcript ID
elfa1	Zm00001d009868_T004	elfa1	Zm00001d009868_T009	elfa1	Zm00001d009868_T003	ZmeIF4E	Zm00001d041973_T001	eif-6	Zm00001d041682_T001
elfa2	Zm00001d037875_T005	elfa2	Zm00001d037875_T008	elfa1	Zm00001d009868_T012	eif-7	Zm00001d014065_T001	elfa2	Zm00001d037875_T006
elfa3	Zm00001d037873_T011	elfa3	Zm00001d037873_T009	elfa2	Zm00001d037875_T003	elfa1	Zm00001d009868_T002	elfa3	Zm00001d037873_T001
elfa9	Zm00001d046449_T004	elfa10	Zm00001d036904_T002	elfa11	Zm00001d037905_T003	elfa12	Zm00001d037877_T005	elfa7	Zm00001d009870_T013
elfa11	Zm00001d037905_T002	elfa10	Zm00001d036904_T005	elfa12	Zm00001d037877_T008				

## Data Availability

The raw data are publicly available from the National Center for Biotechnology Information (NCBI), under BioProject accession number PRJNA846583. The data presented in this study are openly available in zenodo.org at https://doi.org/10.5281/zenodo.6864346 (accessed 19 July 2022). A preprint of this work was published on the “Preprint.org” website at https://doi.org/10.20944/preprints202201.0464.v1 (accessed on 7 July 2022).

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
