# Peer review of "Differential Expression of Genes between a Tolerant and a Susceptible Maize Line in Response to a Sugarcane Mosaic Virus Infection"

_viruses, 2022, doi:10.3390/v14081803_

Round 1

Reviewer 1 Report

The manuscript focuses on the analysis of the RNAseq data to identify potential genes associated with resistance response against Sugarcane mosaic virus (SCMV) in maize. While the authors provided a thorough and interesting analysis of the large transcriptome data generated from 2 parental lines that showed either resistance or susceptibility to SCMV, and the F1 progeny, the ms lacks some actual validation of some of the conclusions, specifically for the interpretation of the CTEPs.

·      Do the B73 line and the CI-RLI line have similar or close genetic / pedigree background? This may be relevant in some of the interpretations of the data.

·      It is unclear why the authors refer to the CI-RLI lines to as resistant in line 37 and next throughout the paper as “tolerant/resistant”.  Is the phenotype depending on the viral strain?  Figure 1 suggests a resistant phenotype for CI-RLI and a tolerant phenotype for F1?

·      For the candidate genes in section 3.5. did the authors validate the RNAseq results via rt-qPCR, for instance for those transcripts of great interest such as eIF6, and argonaute 1 (lines 270-272) as well as some in line 277-283.

·      One interesting analysis of the authors was to define “clusters of transcript patterns” with the justification that maybe some recessive resistance genes or susceptibility factors would not show any change in gene expression.

o   For the genes listed in line 300-308, what is the expression pattern of those genes in the susceptible lines?

o   In their analysis, the authors are looking for recessive resistance genes, which are often characterized by potential mutations within their sequences that do not affect their normal function but may be sufficient to prevent interaction or recruitment by the virus. it is critical for the authors to pick some of their potential genes of interest based on their CTEPs and sequence them to look for potential sequence polymorphisms when compared to that in the susceptible line.  This is of particular relevance for those identified translation factors, as many genes in this category has been reported to be recessive resistance genes.

o   The idea of the CTEP analysis is based on the basal expression level of genes in the resistant line. One of the identified genes using the CTEP approach was BG3, which in the discussion session the authors identified as b-1-3 glucanase (line 493-495). However, measurable changes of B-1-3 glucanase expression level has been reported in responses to viral infections and also its expression can be reversible and feedback regulated with the callose synthase gene.  This brings to question whether the CTEP analysis is in fact a valid approach to identify relevant genes. This supports again the importance of sequencing some of those genes to reveal potential polymorphism to support claim.  

·      Minor comment:

-       Be consistent throughout the ms, either refer to the resistant lines in both text and figures as CI-RLI or CI-RL or RL but not all three.

-       From a methodical approach, provide more details on how the clusters for the CTEPs were determined and how the groups were determined (line 297-298)

Reviewer 2 Report

The paper reports identification of genes differentially expressed in resistant and susceptible maize lines, as well as their cross line, in response to Sugarcane mosaic virus infection. The obtained RNA-seq data were analyzed using clustering analysis and other bioinformatics methods. Several candidate genes were verified by qPCR.

Although the reported data on their own are not worth much without being confirmed by functional studies, the reported dataset is a good starting point for future identification of genes that determine virus susceptibility/resistance in maize. Therefore, to my mind the paper is worth publishing. However, I have minor concerns, which should be addressed before the paper can be accepted for publication. 

1. The paper title is misleading and should be corrected. The role of the proteins mentioned in the current title, even their potential role, cannot be suggested based on the reported data and should not be mentioned in title. To my mind, the title that correctly reflects the paper content would be: ‘Genes differentially expressed in resistant and susceptible maize lines in response to Sugarcane mosaic virus infection’.

2. The timepoint of plant sampling for RNA-seq (17 dpi) is not justified properly. If an early response to virus is analyzed (what is typically done when plant response to virus infection at the gene expression level is analyzed), samples should be taken at the timepoint when the virus can first be detected in systemic leaves, rather than at the time of symptom manifestation. Therefore, the approach used should be better explained in the paper.

Author Response

Thank you for your comments. Please see the attachment

This manuscript is a resubmission of an earlier submission. The following is a list of the peer review reports and author responses from that submission.

Round 1

Reviewer 1 Report

Rodríguez-Gómez et al. analyzed the host responses against the Sugarcane mosaic virus in three different maize lines via RNA-seq, figuring out that translation and ROS detoxification are vital components. Generally, the manuscript is easy to follow. However, the conclusion of the manuscript was not clearly outlined, though many statements were discussed. Below, please find significant concerns.
1, It is hard to believe that ROS detoxification is a crucial component involved in host defense against viruses. The finding of the genes involved in ROS detoxification may be evidence that the ROS accumulates upon virus infection. If genes in ROS detoxification were found to change significantly, the expression of genes involved in ROS accumulation and generation should also be. Unfortunately, the data was not shown in the results. 
The sentence in lines 286-289 was also confusing, “Thus, this increase appears to be more a consequence of an early response to infection, rather than a strategy used for resistance.” This statement is away against the title.
2, As the central part of the manuscript, the bioinformatics analysis should be treated carefully. In figure 2, DEGs may not be appropriately aligned. For example, in Figure 2A, the differently expressed genes in B73 were 924 (772+99+8+45). However, when summing up the upregulated and downregulated genes in Figures 2B and 2C, the differently expressed genes were 955 (B: 496/C:459). An explanation is needed.
3, In Table 1, the classification of GO annotation/terms for some genes are confusing, too. For instance, the Thioredoxin superfamily proteins are assigned to “Detoxification/Cellular detox,” while the Thioredoxin-like 4/Thioredoxin-like protein CXXS1 are in “Biological regulation.” Similarly, argonaute1b belongs to the “Biological regulation”; argonaute1, however, is in “Gene silencing by miRNA.” Thus, what rules did the authors apply to do the classification? More explanation is needed.
Together with those concerns, I am not convinced that the manuscript is ready for publication.